# From Dysbiosis to Neurodegenerative Diseases through Different Communication Pathways: An Overview

**DOI:** 10.3390/biology12020195

**Published:** 2023-01-28

**Authors:** Giorgia Intili, Letizia Paladino, Francesca Rappa, Giusi Alberti, Alice Plicato, Federica Calabrò, Alberto Fucarino, Francesco Cappello, Fabio Bucchieri, Giovanni Tomasello, Francesco Carini, Alessandro Pitruzzella

**Affiliations:** 1Biomedicine, Neuroscience, and Advanced Diagnostics (BIND), University of Palermo, 90127 Palermo, Italy; 2Euro-Mediterranean Institute of Science and Technology (IEMEST), 90136 Palermo, Italy; 3Universitary Consortium of Caltanissetta, University of Palermo, 93100 Caltanissetta, Italy

**Keywords:** gut microbiota, gut microbiome, dysbiosis, neurodegenerative diseases

## Abstract

**Simple Summary:**

For millennia, humankind has had a symbiotic relationship with different bacterial species, called the microbiota. Our microbiota can affect the homeostasis, the health status and the various functions of our body. In this paper, correlations between gut microbiota (GM) and several neurodegenerative processes will be analyzed. We focus our analysis on the possible alterations to the GM (dysbiosis) and their consequences on the human nervous system. Generally, there is an exchange of molecules between the bacterial and the nervous cell populations. An alteration of this communication can evolve into a range of pathological conditions. The present analysis covers the different altered molecular mechanisms and the pathological states observed in conjunction with them. The knowledge of these mechanisms could lead to the development of new therapeutic targets that could counteract neuroinflammation and neurodegeneration.

**Abstract:**

The microbiome research field has rapidly evolved over the last few decades, becoming a major topic of scientific and public interest. The gut microbiota (GM) is the microbial population living in the gut. The GM has many functions, such as maintaining gut homeostasis and host health, providing defense against enteric pathogens, and involvement in immune system development. Several studies have shown that GM is implicated in dysbiosis and is presumed to contribute to neurodegeneration. This review focuses mainly on describing the connection between the intestinal microbiome alterations (dysbiosis) and the onset of neurodegenerative diseases to explore the mechanisms that link the GM to nervous system health, such as the gut-brain axis, as well as the mitochondrial, the adaptive humoral immunity, and the microvesicular pathways. The gut-brain communication depends on a continuous bidirectional flow of molecular signals exchanged through the neural and the systemic circulation. These pathways represent a possible new therapeutic target against neuroinflammation and neurodegeneration. Progress in this context is desperately needed, considering the severity of most neurodegenerative diseases and the current lack of effective treatments.

## 1. Introduction

In the past few decades, the potential role and the mechanisms of action of the gut microbiota (GM) have aroused considerable interest within the medical and scientific fields. The GM refers to the symbiotic bacteria living in the gut. The GM is involved in numerous functions that are required for the maintenance of intestinal homeostasis and for host health. The microbiota elements are also involved in the metabolic mechanisms involving undigested nutrients, in supplying beneficial microbial metabolites, in the defense mechanism targeting enteric pathogens, and in immune response development [1].

Based on the latest evidence identifying the eukaryotic organisms as meta-organisms, these can now be considered functional units together with their microbiota [2]. The correlation between the individual alimentary habits and the correct development of the microbiotic component of this unit has been well established. The dietary habits of an individual have been shown to influence microbial colonization, maturation, and changes throughout the individual’s lifespan [3].

The most recent studies have shown that the GM also affects brain physiology and pathology. In regard to the latter, gut dysbiosis is presumed to contribute to neurodegeneration. These pathogenetic changes occur through various signaling pathways, such as those activated during nerve inflammation, oxidative stress, energetic metabolism, and mitochondrial function (Figure 1). Therefore, dietary habits throughout an individual’s lifetime may contribute to the onset of intestinal dysbiosis, which, in turn, could influence the onset of neurodegenerative diseases.

To better depict the GM in its microenvironment and its relevance in understanding the intestinal and extraintestinal manifestations of GM imbalance, we have recently coined a novel term, the muco-microbiotic (MuMi) layer, which refers to a set of three elements: the GM; the mucus (produced by the intestinal epithelial cells, IECs) in which the GM grow and proliferate; and the nanovesicles (i.e., the exosomes produced by the human cells and the membrane vesicles produced by the GM) regulating the trafficking between the host and the GM elements [4,5].

This review aims to examine the mechanisms linking the GM to brain health, focusing on the correlation between alterations of the intestinal microbiota (*dysbiosis*) and the onset of the neurodegenerative diseases. This connection involves several communication pathways: the gut-brain axis, the mitochondrial pathway, the humoral adaptive immunity, and the microvesicular pathways (Figure 1).

## 2. Gut Microbiome and Microbiota

The human intestine is one of the microbial ecosystems with the highest natural population density. An estimated ~100 trillion microorganisms live in the adult intestine, exceeding the total amount of cells in the human body (10^13^) [6,7,8]. The first studies on the microbiome date back to the 1980s, when Whipps and colleagues defined its characteristics.

The term microbiome was first coined by combining the terms “micro” and “biome” to identify a specific microbial population in a clearly defined environment that is characterized by distinctive physicochemical properties. Although many scientists today refer to the human microbiome as our “last organ” (meaning the most recently discovered one) [9], we disagree with this opinion because, as previously stated, it would be better considered as a part of the innermost layer of the intestine, the muco-microbiotic (MuMi) layer, i.e., the layer made not only by the microbiota but also by a mucous matrix and by nanovesicles (which are very important for cell cross-talk) that are secreted by both human and microbiotic cellular elements [4,5,10].

The microbiome research field has grown rapidly in recent decades, becoming one of the key topics of scientific interest [11]. 

Historically, the main focus of this topic has been environmental microbiome research, such as microbial ecology, which provides an interdisciplinary platform for many disciplines, such as agriculture, food science, biotechnology, and, in particular, human medicine.

Broadly speaking, the human GM contains organisms from all life domains. The phylotype composition of the GM varies depending on several factors, such as diet, environment, and stress [12], but also on concurrent intestinal infections or diseases and antibiotic treatments [13]. Colonization of the gut leads to the establishment of a commensal and symbiotic host-microbiota relationship determined by the production and release of multiple bioactive metabolites. 

This complex interaction sees the gut microbiota play a major role in regulating several mechanisms involved in maintaining the correct metabolic, trophic, and protective functions of the intestine [14,15]. For example, the intestinal microbiota drives the normal intestinal development, acting on the proliferation and apoptosis of the IECs [16,17].

## 3. Gut Dysbiosis

A decrease in the number of microbial species and the loss of several beneficial bacterial strains (Bacteroides such as the Bacteroidiaceae family and Firmicutes) can cause dysbiosis.

Dysbiosis can also be induced by an increase in pathobacteria, such as Prevotellaceae, Akkermansiaceae, and Enterobacteriaceae, like Escherichia coli. 

The connection between dysbiosis and metabolic disorders is well known [18,19], and can compromise the host health by promoting endogenous intoxication, systemic inflammation, and reducing essential metabolites. 

Altered bacterial flora can impair the intestinal function, causing various disorders, ranging from abdominal pain and bloating, accompanied by general malaise, constipation, and colitis, to the onset of bacterial infections, reflux, allergies, autoimmune diseases, and colon cancer [20].

Several causes of dysbiosis have been recognized, such as an unhealthy diet rich in refined foods, excessive consumption of alcohol and hard liquors, additives and pollutants, drugs (e.g., antibiotics), stress, and the after-effects of bacterial and/or viral infections. There is also a high risk of a marked and chronic gut dysbiosis following gastrointestinal surgeries and treatment with antibiotics and other drugs.

## 4. Dysbiosis and Neurodegeneration

Neurodegeneration can be caused by the microbiota through various mechanisms (Figure 1). Emerging studies suggest that the gut dysbiosis may influence the development and evolution of various neurological disorders. The gut dysbiosis intervenes in the development of local and systemic inflammatory states, resulting in the altered integrity of the intestinal epithelial barrier (IEB) [21]. At the same time, dysbiosis causes increased permeability of the brain parenchyma. This dysfunction could lead to neuroinflammation and neuronal cell malfunction. The release of bacterial endotoxins, such as lipopolysaccharide (LPS), is a consequence [22].

A long-standing study reported higher levels of LPS and other Gram-negative bacterial molecules in the human brains affected by Alzheimer’s disease (AD), compared to the healthy controls. The increased LPS levels colocalized with β-amyloid (Aβ) deposits in the plaques, suggesting possible interactions between LPS and Aβ [23].

Additionally, LPS increases β-amyloid fibrillation in vitro [24] and potentially triggers nuclear factor kappa signaling that, in turn, activates B-cells (NF-kB) at the neuronal level, causing nerve inflammation [25].

Furthermore, LPS can affect Aβ transport by increasing its uptake across the blood-brain barrier (BBB) [26]. It has also been observed that chronic systemic inoculations of lipopolysaccharide for up to one week promote various cognitive dysfunctions, as well as β-amyloid plaque production in the hippocampus, and microglial activation in wild-type (WT) mice.

The neurodegenerative processes can be aggravated by peripheral stimulation [27]. Recently, Aβ has been observed in the gut, potentially triggering or enhancing Aβ aggregation in the brain. Amyloid peptides produced by different bacterial families, such as Akkermansiaceae and Prevotellaceae, and prokaryotic families, such as Enterobacteriaceae and Bacteroidiaceae, can act as scaffolds for structural integrity and biofilm production. Conversely, beneficial prokaryotic families (e.g., Bifidobacteriaceae and Lachnospiraceae) decrease in AD patients (Table 1 and Figure 2).

Moreover, these molecules are very similar to Aβ particles in the brain and are recognized by toll-like receptor 2 (TLR2) that are expressed by inflammatory cells, triggering intestinal inflammation [28]. The presence of impaired the IEB and the BBB in a dysbiotic environment leads the amyloid peptides to enter the bloodstream and enhance the aggregation of amyloid plaques. Another viable theory is that endogenous Aβ production occurs in the gut and, at a later stage, spreads to the central nervous system (CNS).

Indeed, increased levels of Aβ have been detected in the intestinal mucosa of AD animal models [28] and patients [29]. Furthermore, reliable evidence supports the role of dysbiosis in Parkinson’s disease (PD)-related neurodegeneration. A greater number of bacterial families (e.g., Verrucomicrobiaceae and Akkermansiaceae) are increased in PD patients. Simultaneously, fewer short-chain fatty acid-producing bacteria (SCFA) were observed [30,31]. Conversely, beneficial prokaryota families (e.g., Lactobacillaceae and Lachnospiraceae) have been shown to be decreased in PD patients, compared to healthy controls (Table 1 and Figure 2). 

Another feature analyzed was the potential long-lasting neurotoxic effects of dysbiosis. The PD-related dysbiosis was connected to significant functional changes in the intestinal lumen, affecting numerous metabolic pathways, such as the metabolic mechanisms of the amino acids, the carbohydrates, and the xenobiotics [32,33]. 

Recently, Kishimoto et al. reported another connection between intestinal inflammation and dopaminergic neurotoxicity, finding that mild chronic intestinal inflammation exacerbates neuropathology and neuromotor deficiency in α-Syn-mutant mice [34]. The presence of Lewy bodies in the submucosal and the myenteric plexuses of the gastrointestinal tract of PD patients was reported more than 30 years ago by Wakabayashi and colleagues [35]. Subsequent studies have shown that the pathological aggregation of α-syn in the intestine occurs in 65–85% of the affected patients [35,36], with a descending frequency gradient, and with the maximum clustering observed in the lower gastrointestinal tract and the stomach. 

Furthermore, pathological α-syn can be detected in the early stages of PD, predating diagnosis by 10–20 years [37,38], and its severity correlates with the severity of constipation and the extent of the enteric neuronal loss in the PD patients [39]. Active axonal transport of α-syn precipitates from the intestine to the brain via the vagus nerve has been evidenced in mice injected with preformed α-syn fibrils in the intestinal wall, resulting in delayed and progressive PD-like neuropathology and some movement disorders [40,41]. Functional analysis revealed alterations in several pathways involving the enzyme, the nucleotide, and the carbohydrate metabolism, as well as the GM composition changes over time [42]. 

Additionally, in Amyotrophic Lateral Sclerosis (ALS) mice, SCFA-producing bacteria (e.g., Akkermansiaceae, Prevotellaceae, and Oscillospiraceae) and prokaryota families, (e.g., Bacteroidiaceae, Lactobacillaceae, and Bifidobacteriaceae) play a significant role in dysbiosis preceding the onset of the movement disorders [43]. Conversely, beneficial prokaryota families (e.g., Peptostreptococcaceae) are decreased in ALS patients (Table 1 and Figure 2). Unlike in other models of neurodegeneration, depletion of the microbiota through altered conditions or antibiotic use has been shown to exacerbate this disease phenotype [44]. In addition, Burberry et al. demonstrated the fundamental role of C9orf72 in modulating the inflammatory response of the vagus nerve triggered by the GM alterations to the CNS X. C9orf72 is the most common genetic variant that contributes to ALS., Specifically, C9orf72 prevents the microbiota from inducing a pathological inflammatory response. It has been shown that in C9orf72 knockout mice, a microbial environment enriched with pro-inflammatory bacteria causes exaggerated systemic and neural inflammation [45]. 

This review summarizes different interaction mechanisms between the microbiota and the host and how dysbiosis could influence the neurodegeneration processes.

**Table 1 biology-12-00195-t001:** Several bacterial and prokaryotic phyla are implicated in dysbiotic processes related to the onset of neurodegenerative diseases. The Akkermansiaceae and Prevotellaceae families are increased in Alzheimer’s, Parkinson’s, and ALS patients. Likewise, prokaryotic Bacteroidiaceae families are present in AD and ALS. In contrast, the Lachnospiraceae and Peptostreptococcaeae are respectively less numerous in these diseases. Opposite patterns of bacterial abundance were found in the Bifidobacteriaceae families between AD and ALS; in the Oscillospiraceae, and in Lactobacillaceae between PD and ALS [23,30,42].

Neurodegenerative Disease	Kingdom	Phylum	Class	Family	Alteration of Abundance	References
**Alzheimer’s** **disease**	Bacteria	Verrucomicrobiota Bacteroidota Proteobacteria Bacteroidota Actinobacteria Firmicutes	Verrucomicrobiae Bacteroidia Gammaproteocbacteria Bacteroidia Actinobacteria Clostridia	Akkermansiaceae Prevotallaceae Enterobacteriaceae Bacteroidiaceae Bifidobacteriacedae Lachnospiraceae	+ + + + − −	[23,25,27]
**Amyotrophic lateral** **sclerosis**	Bacteria	Verrucomicrobiota Bacteroidota Actinomycetota Firmicutes Bacteroidota Firmicutes Actinobacteria Firmicutes	Verrucomicrobiae Bacteroidia Coriobacteria Clostridia Bacteroidia Bacilli Actinobacteria Clostridia	Akkermansiaceae Prevotellaceae Coriobacteriaceae Oscillospiraceae Bacteroidiaceae Lactobacillaceae Bifidobacteriaceae Peptostreptococcaeae	+ + + + + + + −	[42,43]
**Parkinson’s** **disease**	Bacteria	Verrucomicrobiota Bacteroidota Firmicutes Firmicutes Firmicutes Firmicutes	Verrucomicrobiae Bacteroidia Clostridia Bacilli Clostridia Clostridia	Akkermansiaceae Prevotellaceae Oscillospiraceae Lactobacillaceae Lachnospiraceae Clostridiaceae	+ + − − − −	[30,31,32]

## 5. Microbiota-Gut-Brain Axis

The microbiota-gut-brain axis consists of a series of neural connections involving the CNS, the autonomous nervous system (ANS), and the enteric nervous system (ENS). The function of this framework is based on neural and humoral signaling.

A constant flow of direct and indirect signals transmitted through the immunological system, the neurological pathways, and the systemic circulation are involved in the gut-brain communication. 

In addition, the ENS communicates with the brain through afferent neural circuits composed of sensory nerves, through which modulatory stimuli are conveyed in order to generate gut reflexes. Moreover, enteric communication integrates the modulation of immune activity through the receptors expressed by the immune cells.

The brain’s perception of the intestinal environment is facilitated by the vagus nerve, which is composed of 80% visceral afferent fibers. Neurotransmitters, such as serotonin, dopamine, and ɣ-aminobutyric acid (GABA), are crucial in this communication process. Indeed, these chemical mediators act by sending the endogenous impulses through the vagus nerve and the sympathetic and the parasympathetic pathways [46]. Furthermore, the pathogenic bacteria induce inflammatory states in the gut through the production of a large amount of the hormones, the peptides, and the microbial metabolites, such as the SCFAs, the secondary bile acids, and the products derived from tryptophan and polyphenols. This has a significant impact on the development of the neuronal structure and the triggering of the neurodegeneration processes (Figure 1a). This infection is perceived by the brain through an early warning signal sent after an activation of the vagal sensory ganglia and the nucleus tractus solitarii [47].

In fact, in response to the afferent signals, the CNS communicates with the gut through different signaling pathways that are critical for the regulation of motility, mucus secretion, barrier integrity, and visceral sensitivity [48]. 

In addition, emotionality and fear reactivity in infants have been shown to be dependent on the GM composition [45]; this is important evidence for aiming to predict the risk of anxiety and depression. Furthermore, the gut bacteria can influence the human personality; for example, sociability may be associated with a better GM diversity, whereas anxiety and stress are correlated with reduced diversity [49]. The microbiota-brain-gut axis communicates through the humoral signaling molecules and the hormonal components, and not only through the classical neural pathways. Impaired gastrointestinal and brain function may be a consequence of this complex network’s actions [50].

The signal flow passing between the gut and the brain includes indications of harmful stimuli, such as intestinal distension or potential danger signals, including the presence of bacterial endotoxins or pro-inflammatory cytokines. 

The brain receiving this information can lead to changes in gut physiology or immune function (e.g., cytokine secretion) [51].

## 6. Adaptive Humoral Immunity Pathway

The adaptive humoral immunity pathway consists of the hypothalamus-pituitary-adrenal complex, the entero-endocrine system, and the immune system at the level of the intestinal mucosa.

The humoral immunity response (HIR) is facilitated by the release of the antibodies produced by the B lymphocytes that differentiate into the plasma cells. These antibodies bind to the surface of the antigens, such as those presented by viruses, bacteria, and other non-self-molecules, and elicit an immune response. Furthermore, the complex HIR system is known to be one of the possible pathways by which gut dysbiosis affects the brain, leading to the development of neurodegenerative diseases.

It is crucial to emphasize the important significance of the cooperative state in which the microbiota and the innate mucosal immune system co-exist. In particular, an adaptive multitude of immune cell populations, including IgA-producing plasma cells, γδT cells, and CD4+ T cells dominated by a Th1 or Th2 phenotype, constitute the elaborate gut wall immune system [52]. Alteration of the pro-inflammatory cytokines profile through toll-like receptor activation is a consequence. 

In fact, recent studies have shown that the CD4+ T-cell pool of the intestinal mucosa contains many Th-17 cells, which produce Interleukin-17 and T-reg cells with regulatory function. In addition, the presence of IL-22-producing NK-22 cells has also been reported (Figure 1f) [53].

## 7. Mitochondrial and Lysosomal Pathway 

A growing body of research highlights the bidirectional flow of information between the GM and the mitochondria [54]. Since the nervous system has a high metabolic rate, the brain is inevitably affected whenever abnormalities in mitochondrial function occur, leading to dysfunctions that can result in diseases. 

Extensive studies document a close connection between the gut-brain axis and mitochondrial function [55]. Human cells contain hundreds to thousands of mitochondria, which vary in shape, number, and size, depending on several factors.

The oxidation of sugars and fats occurs within the mitochondria to produce adenosine triphosphate (ATP). The two essential metabolic pathways are the mitochondrial oxidative phosphorylation, consisting of electron transport through cellular respiration, and the β-oxidation of fatty acids [56]. Interestingly, the bacteria and the host mitochondria share many common features. 

According to the endosymbiotic theory, the ancestor of mitochondria was a member of the phylum alpha-proteobacteria that formed a symbiotic association with the eukaryotic cell, giving it structural features and functions similar to those of bacteria [57]. Moreover, the GM influences neurodegeneration altering mitochondrial activities. 

In particular, the role of SCFAs, reactive oxygen species (ROS), nitric oxide (NO), and hydrogen sulfide (H2S) in the communication between the microbiota and the mitochondria host are well known (Figure 1c). The influence of microbiota on the brain is exploited by several molecular mechanisms. Many of these processes are involved in neurodegenerative pathogenesis [57]. For example, the action of SCFAs is relevant in maintaining the integrity of the BBB structure, a crucial element in brain development and homeostasis. Mitochondrial function and dysfunction manage the regulation of butyrate and, therefore, its effects. Mitochondrial oxidative phosphorylation is maintained by butyrate, and the consequence that has been noted is immune reactivity. Scientific evidence supports the neuroactive properties of SCFAs crossing the CNS [58], although their mechanisms of action in this environment remain uncertain. However, in the animal models, they have been postulated to influence important neurological processes and be involved in the onset of neurodegenerative disorders [59].

Beneficial effects of mitochondrial activity that are lost due to gut dysbiosis include a decrease in butyrate levels. 

Moreover, mitochondria release toxic ROS in response to an elevated metabolism, hypoxia, or mitochondrial damage (Figure 1c) [60]. Alterations in the mitochondrial function cause genetic changes and affect extracellular signaling through the release of extracellular vesicles (e.g., exosomes) [61]. ROS has been identified as a key molecule in the pathogenesis of AD [14], PD [21], Huntington’s disease (HD) [22], ALS [23], and Multiple sclerosis (MS) [21].

The oxidative state of the brain can be regulated by the GM, at the level of the ROS and antioxidant system, through the production of SCFAs, which control the permeability of the intestinal barrier and the modulation of the immune system, as well as influencing mitochondrial function through reduced ROS production. 

However, mitochondrial dysfunction could occur if there is a decrease in the butyrate levels and the subsequent production of ROS [62]. It has been observed that NO generated by the gut bacteria near the intestinal mucosa may exert beneficial effects at physiological levels (Figure 1c). 

However, the overproduction of NO interferes with standard functions. Therefore, bacterial NO formation in the gut can be considered a modulator of the physiological and the pathological effects [63]. 

Physiologically, NO signaling links cellular energy demand with the mitochondrial energy supply. This pattern positively influences the mitochondrial oxidative state [64]. Specifically, NO binds to the human proteins in a carefully regulated manner -a process known as S-nitrosylation- that can lead to increased mitochondrial fission and reduced mitochondrial network size, as well as the cellular ability to generate energy when produced at high levels. NO signaling is used by intestinal bacteria to communicate with the hosts, and it is a critical regulator of key intestinal processes. Dysregulation of this mechanism is widely implicated in neurodegenerative diseases [65]. Hydrogen sulfide (H2S) is a molecule produced by epithelial cells and intestinal microbiota from cysteine in the gut (Figure 1c). Low concentrations of H2S in the intestine increase the respiration capacity of colonic mucosal cells. H2S may protect neurons from apoptosis and degeneration [66]. In addition, H2S is closely involved in reducing oxidative stress, preventing oxidative DNA damage and mitochondrial dysfunctions [67]. 

In contrast, excessive concentrations of H2S are harmful, either in combination with a downregulation of the enzymes involved in its mucosal detoxification, or in the presence of dysbiosis [68]. H2S metabolism occurs during the mitochondrial oxidation; H2S inhibits the mitochondrial cytochrome oxidase already at sub-micromolar concentrations, resulting in a blockage of aerobic respiration and oxidative phosphorylation, eventually leading to cell death [69]. 

Therefore, excessive H2S production leads to the activation of neuro-cytotoxic mechanisms in the brain. Intestinal dysbiosis is associated with a reduced wellbeing of the host; SCFAs, ROS, NO, and H2S are essential signaling molecules that can be harmful if they are present in excess. Increased levels of these molecules can affect human health, gut cell homeostasis, and the biodiversity of the gut microbial community. Reciprocally, gut microbes can influence SCFA, ROS, NO, and H2S levels (Figure 1c), mitochondrial homeostasis and host health.

In addition to the mitochondria, the lysosomes are also essential for proper cellular functioning, as they provide a key signaling platform by regulating many crucial processes such as autophagy, proliferation, and cell death. Thus, a malfunction of the cellular surveillance systems inexorably leads to toxicity and, often, to cell death, due to the accumulation of unwanted nonfunctional components within cells [70]. Substantial evidence indicates how dysbiosis can modify and contribute to dysregulation of the autophagy-lysosomal protein clearance mechanism. In this way, autophagosomal-proteasomal pathways participate in a vicious cycle that causes cytotoxicity and aggravates pathologies typical of neurodegenerative disorders, such as AD and PD. In addition, chronic inflammatory diseases, often caused by dysbiosis, are associated with LRRK2 (Leucine-Rich Repeat Kinase 2) gene variants. Although the physiological and pathological impact of LRRK2 is still unclear, mounting evidence supports a role for LRRK2 in membrane and vesicle trafficking, mainly in the endosome recycling system, autophagy, and lysosome biology [71,72]. 

Therefore, dysbiosis may underlie the susceptibility of many chronic inflammatory diseases, both intestinal and systemic. 

## 8. Microvesicular Trafficking Pathway

In the intestinal system, the development of immune function depends on the elaborate interaction between the non-pathogenic bacteria, the IECs, and the immune cells of the mucosal tissue. This communication occurs not only through direct contact but also through the humoral signaling molecules and the hormonal components. Intercellular communication with the host also occurs through micro- and nano-vesicles that can enter the systemic circulation (Figure 1e). 

In particular, the maintenance of the tissue homeostasis is mediated by the production of the extracellular vesicles (EVs), secreted by all cell types, contributing significantly to coordinated signaling events and crosstalk between the microbiota, the IECs, the endothelial cells, and the immune cells [73]. In addition to the lipids, the proteins, and the miRNAs, also metalloproteinases, growth factors, and chemokines, which are used as secondary messengers for coordinating the cellular response, are transported within the EVs [74]. 

In general, the IEC-derived EVs act on the regulatory process of the epithelial barrier integrity by transporting the desmosome cadherins that stabilize the epithelial cell-cell adhesions. In addition, these EVs can protect against pathogenic infections by transporting certain antimicrobial peptides, such as beta-defensin [75,76]. 

Conversely, Gram-negative and positive bacteria and the eukaryotic cells can release EVs from their membranes, called the membrane vesicles (MVs) or the outer membrane vesicles (OMVs) (Figure 1e), respectively. These spherical particles, varying in diameter (20–200 nm), are produced by the leakage of the outer membrane. They contain a variety of molecules, such as the DNA, the RNA, the lipids, and the enzymes [77,78]. Like the exosomes, the microbiota-derived EVs can travel a long distance, transporting their contents throughout the body and facilitating cell-to-cell communication and target cell activity. This feature suggests that the OMVs may also act in other districts besides the intestinal lumen, amplifying the role of the microbiota (Figure 1e). 

In fact, innovative studies have shown that communication with the host also occurs through the release of the microvesicles that can enter the systemic circulation and cross the blood-brain barrier (Figure 1d) [79].

The microbiota can establish direct contact with other compartments of the body through the intestinal wall. The elaborate communication network between the gut and the CNS is strengthened as a result. 

Recently, the role of the OMVs produced by the intestinal microbiota has been the subject of numerous studies, which have highlighted their importance as immunological mediators. It has been shown that the capsular polysaccharide (PSA) is selectively packaged within the OMVs. The OMVs are then internalized into the dendritic cells, which induce the differentiation of the T-regulatory cells to produce IL-10 (Figure 1e). This pathway represents an important mechanism of host immunotolerance against the symbiont [80,81].

Other studies have shown that the OMVs can benefit the entire intestinal microbial population. Some Bacteroides can package hydrolases within the OMVs, making them available to other bacteria that lack them. This mechanism promotes the growth of other bacterial species that cannot hydrolyze the polysaccharides on their own. Again, this supports the role of the OMVs in creating and maintaining the balance of the GM [82].

## 9. Endotoxin Pathway

The endotoxin theory of neurodegeneration suggests their involvement in the neurodegeneration process [83]. The endotoxin lipopolysaccharide (LPS) forms the outer membrane of Gram-negative bacteria and is highly expressed in the intestine, the gums, the skin, and other tissues during bacterial infections. The standard amount of the endotoxin in the plasma is typically low; however, it increases during contamination, intestinal irritation, gum disease, and neurodegenerative infections. The introduction of the endotoxin into the blood of healthy subjects causes fundamental irritation and the onset of microglia in the brain. This effect suggests that the endotoxin synergizes with various grouped proteins to trigger different neurodegenerative diseases. Indeed, high levels of endotoxin cause the accumulation of amyloid β, tau, and α-synuclein, whose molecular interactions modulate neuropathology (Figure 1b) [84]. 

In addition, the binding of endotoxins to Apolipoprotein E (APOE) causes a variation in APOE4 and exacerbates the endotoxin production, promoting susceptibility to Alzheimer’s disease. 

Alterations in the microbiota and subsequent intestinal permeability in PD induce microgliosis by LPS, which is associated with the metabolic disturbances characterizing the disease and leading to the progressive deterioration of motor function. This confirms that bacterial species driving endotoxin changes may influence infection in patients with Parkinson’s and Alzheimer’s disease [84]. Although the mechanism is not yet well understood, it is likely that the phagocytic internalization of the bacterial LPS acts on the protein function rather than the protein expression, thus influencing the process of aggregation and fibrillation. 

Therefore, the LPS could trigger a self-amplifying impact on the aggregation kinetics of α-synuclein and amyloid β in the early stages of PD and AD, respectively. Preliminary studies have thus reported that the combination of these aggregated proteins with inflammatory mediators could, in turn, affect the CNS by influencing the disease onset and progression [85,86]. 

Other recent studies showed increased endotoxin levels with amyotrophic sclerosis, resulting in increased total TDP-43 and neuropathology. In this case, it would appear that the endotoxin implicitly drives the microglia to cause neuron damage through the release of nitric oxide, oxidants, and pro-inflammatory cytokines. 

Hypotheses on the possible involvement of the endotoxin in this mechanism are still debated. However, a reduction in the degree of neurodegeneration could be the consequence of reduced endotoxin expression or endotoxin-induced neuroinflammation.

## 10. Conclusions

The GM is known to exert significant effects on the entire organism, particularly on the brain. This review highlights how dysbiosis is correlated with the onset of neurodegenerative diseases, such as AD, PD, and ALS. The different pathways through which gut dysbiosis induces neurodegeneration were also discussed. 

The signal flow passing through these pathways includes several potentially harmful molecules, such as the endotoxins and the pro-inflammatory cytokines. The onset of these signals appears to be linked to the development of dysbiosis-mediated neurodegenerative diseases. 

The imbalance between the intestinal microbiota and the innate immune system of the mucosa is highly significant, and it appears to alter the profile of the toll-like receptor-mediated pro-inflammatory cytokines. 

In addition, the mitochondrial pathway was investigated, focusing on SCFAs, ROS, NO, and H2S, which play a crucial role in the crosstalk between the microbiota and the mitochondria of the host. 

Furthermore, the OMV production by the different bacterial subgroups was considered a possible immunological mediator. In conclusion, the endotoxins that drive the microglia and induce the neuronal damage using nitric oxide, oxidants, and cytokines, were discussed. 

The analyzed pathways may provide possible new therapeutic targets against neuroinflammation and neurodegeneration. Any progress in this direction is desperately needed, considering the severity of most neurodegenerative diseases and the current lack of effective treatments. Therefore, focusing on the GM healthiness could be relevant to properly understanding the different neurodegeneration causes. The newly acquired knowledge could be used to act against the major development of neurodegenerative disease. For example, therapeutic protocols could be developed to restore the patient’s microbiota in dysbiosis conditions. Conversely, if the subject has a microbiota predisposing to the onset of pathological conditions, replacement with a selected bacterial population should not be precluded. The same path should be pursued with metabolites involved in the communication networks between the microbiota and the host organisms. The advances made in identifying the key molecules behind this signaling will be available for use in both enhancing the effects of the beneficial molecules and counteracting the pro-pathological effects of the others.

## Figures and Tables

**Figure 1 biology-12-00195-f001:**
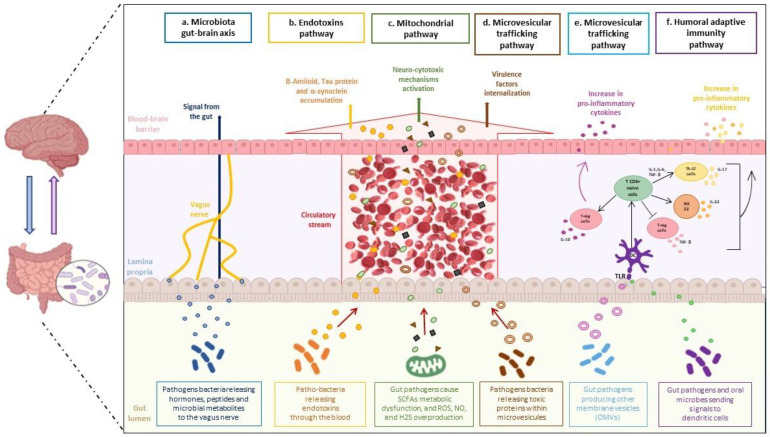
The gut microbiota influences the onset of neurodegenerative diseases. Cooperation between a dysbiotic gut and the central nervous system occurs through several pathways: (**a**) the gut-brain axis, (**b**) endotoxins pathway, (**c**) mitochondrial pathway, (**d**,**e**) microvesicular trafficking pathway, and (**f**) humoral adaptive immunity pathway. The microbiota gut-brain axis (**a**): pathogenic gut bacteria release hormones and microbial metabolites that reach the brain via the vagus nerve, crossing the blood-brain barrier and inducing neurodegeneration processes. Endotoxins pathway (**b**): endotoxins reach the brain through the bloodstream and cause the accumulation of amyloid β, Tau-protein, and α-synuclein, modulating neuropathology. Mitochondrial pathway (**c**): intestinal pathogens cause SCFAs metabolic dysfunction and increased levels of ROS, NO, and H2S; these altered molecules reach the brain through the systemic circulation and induce activation of neuro cytotoxic mechanisms. Microvesicular trafficking pathway (**d**): pathogenic bacteria release toxic proteins within microvesicles that cross the blood-brain barrier through the bloodstream; this event allows internalization of virulence factors; (**e**) Gram-negative and positive bacteria produce vesicles from their membranes; these vesicles are internalized into dendritic cells and induce the differentiation of T-regulatory cells to produce IL-10, to create a neuroinflammatory environment. Humoral adaptive immunity pathway (**f**): gut microbes internalize pathogenic signals to dendritic cells; differentiation of Th-17 cells and NK-22 cells, and the pro-inflammatory cytokines, as IL-17 and IL-22,occur accordingly.

**Figure 2 biology-12-00195-f002:**
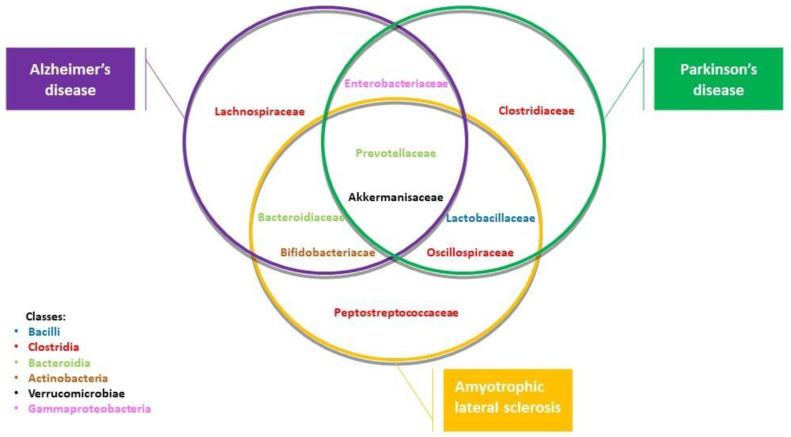
Illustration of bacteria and prokaryotic domains implicated in Alzheimer’s disease (AD), Parkinson’s disease (PD), and amyotrophic lateral sclerosis (ALS). Note how the Akkermansiaceae and Prevotellaceae families are both implicated in these three diseases; the Lactobacillaceae and Oscillospiraceae families are involved in PD and ALS; the Bacteroidiceae and Bifidobacteriaceae families are implicated in AD and ALS, while the Enterobacteriaceae family is involved in AD and PD. The Lachnospiraceae, Peptostreptococcaceae and Clostridiaceae families are involved in AD, ALS, and PD.

## Data Availability

Not applicable.

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
