# Peer review of "From Dysbiosis to Neurodegenerative Diseases through Different Communication Pathways: An Overview"

_biology, 2023, doi:10.3390/biology12020195_

Round 1

Reviewer 1 Report

The manuscript by Intili et al. reviews the connection between intestinal dysbiosis, i.e. imbalance of the gut microbiota, and the development of neurodegenerative disease. The authors describe the different pathways providing crosstalk between the gut and the gut microbiota, on the one hand, and the brain, on the other hand. The manuscript collects many evidences that link dysbiosis with neuroinflammation throughout different pathways (e.g. humoral adaptive immunity, mictochondria, etc).

I have three major concerns:

(1) A figure linking all the pathways and showing interactions among pathways is a must in this kind of article. Otherwise the reader feels that the evidence provided is fragmentary and disconnected.

(2) Many data is available on the genetics of neurodegenerative disorders, particularly Parkinson's and Alzheimer's diseases. But no discussion on this and how dysbiosis could promote/accelerate/advance neurodegeneration is really forthcoming, aside from some reflections on how LPS appears in amyloid brain plaques. Surely more can be said on this topic.

(3) The major genetic cause for neurodegenerative disease is functional impairment of lysosomes. Indeed, hetorozygouse mutations at some lysosomal disease genes underlie most of the cases of Parkinson's. Yet no mention to lysosomal malfunctioning and dysbiosis appears, although a full section is devoted to microvesicular traffic (which would be also impacted by lysosomal dysfunction). This is a glaring omisssion. Lysosomes are also connected to mitochondrial positioning within the cell and are the cellular major nutrient sensor, thus I believe the authors should consider studying those interactions as well.

Author Response

Reviewer #1 Comment #1 (R#1C#1): The manuscript by Intili et al. reviews the connection between intestinal dysbiosis, i.e. imbalance of the gut microbiota, and the development of neurodegenerative disease. The authors describe the different pathways providing crosstalk between the gut and the gut microbiota, on the one hand, and the brain, on the other hand. The manuscript collects many evidences that link dysbiosis with neuroinflammation throughout different pathways (e.g. humoral adaptive immunity, mictochondria, etc). I have three major concerns: A figure linking all the pathways and showing interactions among pathways is a must in this kind of article. Otherwise the reader feels that the evidence provided is fragmentary and disconnected.

Author Reply (AR): We thank the reviewer for providing us with valuable insights to improve our review. Therefore, we have modified the first image, creating a new one that best summarised what was already stated in the text.

R#1C#2: Many data is available on the genetics of neurodegenerative disorders, particularly Parkinson's and Alzheimer's diseases. But no discussion on this and how dysbiosis could promote/accelerate/advance neurodegeneration is really forthcoming, aside from some reflections on how LPS appears in amyloid brain plaques. Surely more can be said on this topic.

AR: We modified paragraph 9 to include some discussion on these topics.

R#1C#3: The major genetic cause for neurodegenerative disease is functional impairment of lysosomes. Indeed, hetorozygouse mutations at some lysosomal disease genes underlie most of the cases of Parkinson's. Yet no mention to lysosomal malfunctioning and dysbiosis appears, although a full section is devoted to microvesicular traffic (which would be also impacted by lysosomal dysfunction). This is a glaring omisssion. Lysosomes are also connected to mitochondrial positioning within the cell and are the cellular major nutrient sensor, thus I believe the authors should consider studying those interactions as well.

AR: We modified paragraph 7 accordingly.

Reviewer 2 Report

This review deals with an interesting topic but fails to advance the field.

It lacks critical discussion, identification of methodological problems and clear identification of future directions for both the research and therapeutic interventions.

The manuscript suffers from writing deficiencies. I am not convinced that all the authors (n=12) read the it prior to submission, as there are numerous errors in the text and even in abstract.

Each individual section of the manuscript should be better structured. The text does not read smoothly, lacks precision and in some sections (for instance on page 7) looks more like a draft with points to be further developed and merged. In addition, there are redundant sentences (for example two consecutive sentences on Lewy bodies), unclear phrases (for example ‘the ratio between the phylum of Bacteroidetes’), scientifically incorrect statements (for example ‘organisms that mutates into pathogenic bacteria’), unexplained abbreviations (for example ‘ALS’) and details whose importance cannot be understood without additional information (for example ‘C9orf72’).

Figure 2 is an oversimplification of the gut microbiota, and does not add much to the understanding of the topic. It remains unclear how the choice of presented taxa was made. On the other hand, appropriate illustrations and tables showing implication of different bacteria in different routes (with possible overlaps) of neurodegenerative processes are lacking.

Author Response

Reviewer #2 Comment #1 (R#2C#1): This review deals with an interesting topic but fails to advance the field. It lacks critical discussion, identification of methodological problems and clear identification of future directions for both the research and therapeutic interventions.

The manuscript suffers from writing deficiencies. I am not convinced that all the authors (n=12) read the it prior to submission, as there are numerous errors in the text and even in abstract. Each individual section of the manuscript should be better structured. The text does not read smoothly, lacks precision and in some sections (for instance on page 7) looks more like a draft with points to be further developed and merged. In addition, there are redundant sentences (for example two consecutive sentences on Lewy bodies), unclear phrases (for example ‘the ratio between the phylum of Bacteroidetes’), scientifically incorrect statements (for example ‘organisms that mutates into pathogenic bacteria’), unexplained abbreviations (for example ‘ALS’) and details whose importance cannot be understood without additional information (for example ‘C9orf72’).

Author Reply (AR): We have taken into account the constructive criticism provided by this reviewer and modified the text accordingly.

R#2C#2: Figure 2 is an oversimplification of the gut microbiota, and does not add much to the understanding of the topic. It remains unclear how the choice of presented taxa was made. On the other hand, appropriate illustrations and tables showing implication of different bacteria in different routes (with possible overlaps) of neurodegenerative processes are lacking.

AR: We have modified figure 1 and table 1, and we added figure 2 to reflect what was suggested by the reviewer. 

Reviewer 3 Report

Dear authors,

I read your review with great interest, which proves once again that we are what we eat. I believe that your review summarizes current information in a logical and easily accessible way for the reader. The bibliography is recent.

I couldn't find what comments to make to this review and I think it can be published as it is.

Author Response

Reviewer #3 Comment #1 (R#3C#1): I read your review with great interest, which proves once again that we are what we eat. I believe that your review summarizes current information in a logical and easily accessible way for the reader. The bibliography is recent. I couldn't find what comments to make to this review and I think it can be published as it is.

Author Replay (AR): We would like to sincerely thank this reviewer for the positive comments and enthusiasm.

Round 2

Reviewer 1 Report

The authors have answered my concerns in their point-by-point response.

Author Response

We are glad that we have been able to correctly answer all of your questions and make the appropriate improvements.

Reviewer 2 Report

Newly added Table 1 and Figure 1 provide an oversimplified, incomplete and inaccurate information. The data presented at the family level do not properly reflect changes in taxa abundances observed in cited studies and several important studies on the topic have not been considered. Moreover, considering separately “bacterial and prokaryotic domains” is unjustified and confusing.

- Bacillota and Firmicutes are synonyms, but a reader would not understand that from data presented in Table 1.

- The estimation of bacterial and human cells cited in the ms is outdated.

-  Debating on whether the microbiome is “last tissue” vs “last organ” is out of the scope of this review

Author Response

  • Newly added Table 1 and Figure 1 provide an oversimplified, incomplete and inaccurate information. The data presented at the family level do not properly reflect changes in taxa abundances observed in cited studies and several important studies on the topic have not been considered. Moreover, considering separately “bacterial and prokaryotic domains” is unjustified and confusing.
  • Thank you for this additional criticism. 

    Regarding Figure 1: we know that it is possible to increase the complexity of the figure itself. But we feel that this may make the image, at the same time, complicated to understand. We would like to keep the figure as simple as possible in order to summarize all the issues discussed in the various paragraphs.

     Regarding Table 1, we have accepted the suggestion and made it simpler to understand.   

  • Bacillota and Firmicutes are synonyms, but a reader would not understand that from data presented in Table 1
  • Thank you for your suggestion. We have accepted it, and we have unified the two phyla.

  • The estimation of bacterial and human cells cited in the ms is outdated.

  • We are grateful for your advice. We have modified the estimation referring to more recent informations.
  • Debating on whether the microbiome is “last tissue” vs “last organ” is out of the scope of this review.

  • Thank you for this additional criticism. This debate is not included in the conclusions. It is mentioned only a few times within the text. The sole purpose of this citation is to emphasize current knowledge about human microbiota.

    Our only scope is to spotlight the dysbiosis condition and how helpful its study can be for therapeutic purposes